# New Potential Agents for Malignant Melanoma Treatment—Most Recent Studies 2020–2022

**DOI:** 10.3390/ijms23116084

**Published:** 2022-05-29

**Authors:** Paweł Kozyra, Danuta Krasowska, Monika Pitucha

**Affiliations:** 1Independent Radiopharmacy Unit, Faculty of Pharmacy, Medical University of Lublin, 20-093 Lublin, Poland; monika.pitucha@umlub.pl; 2Department of Dermatology, Venerology and Pediatric Dermatology, Medical University of Lublin, 20-081 Lublin, Poland; dana.krasowska@gmail.com

**Keywords:** melanoma, targeted therapy, anticancer activity, ion channels, BRAF kinase, MAPK pathway, PI3K–AKT pathway, heterocyclic derivatives, cancer

## Abstract

Malignant melanoma (MM) is the most lethal skin cancer. Despite a 4% reduction in mortality over the past few years, an increasing number of new diagnosed cases appear each year. Long-term therapy and the development of resistance to the drugs used drive the search for more and more new agents with anti-melanoma activity. This review focuses on the most recent synthesized anti-melanoma agents from 2020–2022. For selected agents, apart from the analysis of biological activity, the structure–activity relationship (SAR) is also discussed. To the best of our knowledge, the following literature review delivers the latest achievements in the field of new anti-melanoma agents.

## 1. Introduction

Malignant melanoma (MM) is the most lethal of all skin cancers, although it accounts for only 1% of all skin cancers [1]. Its incidence has increased faster than almost any other cancer in the last 50 years [2]. The American Cancer Society forecasts that by 2022 there will be nearly 100,000 new cases and more than 7500 deaths from MM in the United States alone [3]. The poor prognosis is that the five-year survival rate for patients with stage IV melanoma is only 10% [4]. MM is derived from melanocytes, which are responsible for the production and secretion of the melanin pigment [5]. Overexposure to UV light or radiation leads to damage to the DNA of melanocytes, resulting in uncontrolled proliferation and oncogenesis. Other factors determining the development of this neoplasm include genetic factors, the previous presence of MM, family history and skin phenotype.

The most studied pathway of melanoma development is the mitogen-activated protein kinase (MAPK) pathway, i.e., rat sarcoma virus (RAS)–rapidly accelerated fibrosarcoma (RAF)–mitogen-activated protein kinase (MEK)–extracellular signal-regulated kinase (ERK). The signaling cascade culminates in increased expression of nuclear transcription factors, such as c-MYC and MITF, both responsible for cell proliferation [6]. 

The second most studied pathway is the phosphatidylinositol-3-kinase (PI3K) pathway, i.e., PI3K–protein kinase B (Akt)–mammalian target of rapamycin (mTOR) [7]. It is mainly responsible for anti-apoptotic signaling [8] and is crucial for the development of melanoma resistance to BRAF/MEK inhibitors [9]. The MAPK pathway is more actively related to proliferation and invasion and the PI3K–AKT pathway is mainly responsible for anabolism [6]. RAS can also activate the PI3K cascade [8]. Two signal pathways are shown in a simplified manner in Figure 1. The early stages of melanoma are operable, but in the instance of metastases, the treatment options become limited. In malignant melanoma, chemotherapy provides little benefit. However; in recent years, mortality has decreased by ~4%, which may be related to treatment progress through targeted therapies and immunotherapies [1]. 

A new approach to melanoma therapy is the role of ion channels. The channel of high-voltage activated potassium (BK) and Ca2 + ions is expressed in many different cells [10,11] and modulates various processes by forming molecular signaling complexes [12]. Liu et al. reported that neoplastic cells are characterized by an overexpression of BK channels [13]. A correlation of changes in membrane potential with the cell cycle is observed [14], which is confirmed by many studies [13,15,16,17,18,19,20,21]. In other words, they control cancer proliferation, invasion, and metastasis [14]. Since the key role of ion channels in the development of melanoma has been demonstrated [22], the number of studies devoted to this molecular target has been steadily increasing [23]. They are recognized as new potential targets for new therapies for melanoma [24]. A revolutionary study has been published, exhibiting that the BK activator NS-11021 induces apoptosis and inhibits proliferation in the melanoma cell line IGR39 [25]. Therefore, ion channels may be a new potential molecular target in the treatment of melanoma and their dysregulation may serve as a marker for the prediction of melanoma.

The current trend is to develop new molecularly targeted therapies for specific biomarkers responsible for melanoma progression, which will be an alternative or complement to immunotherapy [1]. Thus, there is a need to search for novel potential agents active against melanoma. 

## 2. The Most Potent Anti-Melanoma Agent from Most Recent Studies 2020–2022

The most active structures with potential activity against melanoma from a recent study are presented below. A short SAR analysis is discussed for the selected structures. New potential derivatives have been classified according to the function of the main ring.

### 2.1. Imidazole Derivatives

Imidazole is a five-membered valuable aromatic heterocyclic moiety, bearing two nitrogen atoms. It exerts a great influence on the anticancer activity and its derivatives showed, among others, properties of inhibitors of tubulin polymerization or kinases [26]. Imidazole derivatives with anti-melanoma activity are presented in Figure 2.

Ali et al. obtained 4-(1H-imidazol-5-yl)pyridin-2-amine derivatives [27]. Authors based their syntheses on the scaffold of Dabrafenib, which is a selective BRAF^V600E^ inhibitor [28]. Over 90% of melanomas contain the BRAF^V600E^ mutation and it is associated with a downregulation of the activity of MEK and ERK effectors [29]. The thiazole ring was replaced with bioisosteric 1H-imidazole, the 2-aminopyrimidine group was questioned with 2-aminopyridine, an ethyl or propyl chain was introduced as an alternative for the 2-fluorophenyl group, the butyl group was replaced with a phenyl ring, a hydroxyl or methyl group was introduced in the 3-position of the phenyl group, and phenyl was introduced as an alternative for the 2,6-difluorophenyl group. Compound **(1)** showed the best activity against melanoma cell lines UACC-62 and M14 with a half-maximal inhibitory concentration (IC_50_ = 1.85 μM and 1.76, respectively), and **(2)** against melanoma cell lines MALME-3M, MDA-MB-435, SK- MEL-28, SK-MEL-5, and UACC-257 with an IC_50_ = 1.51 μM, 1.85 μM, 1.63 μM, 1.54 μM, and 1.99 μM, respectively. Compounds **(1)** and **(2)** exhibited an average IC_50_ = 1.8 µM and 1.88 µM against the aforementioned melanoma cells, respectively. Compound **(3)** possessed the highest activity against BRAF^V600E^ kinase with an IC_50_ = 32.90 ± 1.02 nM [27].

Youssif et al. obtained novel triarylimidazole derivatives [30]. The selection of the triaryl imidazole core was made on previously reported data based on the rational drug design [31,32]. The most promising compound **(4)** possessed the highest activity against melanoma cell line LOX IMV1 with a concentration of 50% of the maximal inhibition of cell proliferation (GI_50_ = 0.17 µM), and inhibited BRAF^V600E^ most effectively with the concentration of a substance that causes the death of 50% of the population (LC_50_ = 0.33 ± 0.10 µM). The authors noted that the activity of arylcarboximidamide derivatives increased in the given series of substituents: 4-chlorophenyl < 2-naphthyl < 4-methoxyphenyl ≤ 1,3-benzodioxole for the most potent agent [30].

Ali et al. obtained novel (1H-imidazol-5-yl)pyrimidine-based derivatives [33]. The starting point for further syntheses was the imidazole derivative SB203580, a specific inhibitor of p38 mitogen-activated protein kinase (p38 MAPK) [34]. The highest activity was exhibited by **(5)** with the final 3-fluorophenylsulfonamide, which was about eight-fold more active with an IC_50_ = 0.9 µM, as compared to standard Staurosporine with an IC_50_ = 7.15 µM against melanoma cell line LOX-IMVI. Compound **(6)**, on the other hand, possessed about 12.5-fold more activity against BRAF^V600E^ kinase with an IC_50_ = 2.49 nM, as compared to the same standard with an IC_50_ = 31.60 nM [33]. 

Ali et al. synthesized novel 4-(imidazol-5-yl)pyridine derivatives [35]. In the current study, the authors focused on optimizing the lipophilic properties and cell permeability while maintaining their enzyme-inhibiting activity. Compound **(7)** exhibited the highest activity against melanoma cell lines LOX IMVI, MALME-3M, M14, MDA-MB-435, SK-MEL-2, SK-MEL-28, and UACC-62 with an IC_50_ = 0.56 µM, 0.67 µM, 0.31 µM, 0.07 µM, 0.54 µM, 0.81 µM, 0.51 µM, and 0.25 µM, respectively, as compared to the standard Sorafenib with an IC_50_ = 1.58 µM, 2.00 µM, 2.00 µM, 1.58 µM, 2.00 µM, 2.51 µM, 1.58 µM, and 1.58 µM, respectively, against the aforementioned melanoma cell lines. Compound **(8)** possessed the highest activity against BRAF^V600E^ kinase with an IC_50_ = 0.530 µM. The derivatives with the most promising activity bore a terminal methyl or isopropyl group, an amide linker, and a hydroxyphenyl moiety [35].

### 2.2. Benzimidazole Derivatives

Benzimidazole is a six-membered benzene ring fused to an imidazole ring at the 4-position and 5-position of the imidazole ring. Its derivatives exert antitumor activity mainly by inhibiting tubulin polymerization, kinase, and topoisomerase inhibitors, apoptosis inducers, and telomerase inhibitors [36]. Benzimidazole derivatives with anti-melanoma activity are presented in Figure 3.

Kong et al. obtained novel 1-(5-(1H-benzo[d]imidazole-2-yl)-2,4-dimethyl-1H-pyrrol-3-yl)ethan-1-one derivatives [37]. The main goal was to find new potential inhibitors of the bromodomain family protein and the beyond-end domain (BET). BETs are responsible for the recruitment of protein complexes to promote the initiation and prolongation of transcription; therefore, their inhibition is a promising cancer therapy strategy. Compound **(9)** possessed the highest activity against melanoma cell line UACC-62 with GI_50_ = 274 nM, as compared to the activity of OTX-015 with GI_50_ = 161.8 nM, which was found in Phase 2 clinical trials as a pan-BET inhibitor [37].

Abdel-Maksoud et al. obtained novel 4-(1H-benzo[d]imidazol-1-yl)pyrimidin-2-amine-linked sulfonamide derivatives [38]. The most active sulfonamide derivative was the pyrimidinylbenzimidazole scaffold bearing a propylamine linker between the sulfonamide and the pyrimidine moiety. Compound **(10)** possessed the best activity against BRAF^WildType^ with an IC_50_ = 0.940 ± 0.065 µM and against BRAF^V600E^ with an IC_50_ = 0.490 ± 0.061 µM, as compared to the reference standard Sorafenib with an IC_50_ = 0.814 ± 0.071 µM, against the latter target. For the given melanoma cell line SK-MEL-5, **(10)** was about four-fold more active with an IC_50_ = 2.02 ± 0.09 μM, as compared to Sorafenib with an IC_50_ = 9.22 ± 0.81 µM, and for the given melanoma cell line A375, two-fold more active with an IC_50_ = 1.85 ± 0.11 µM than Sorafenib with an IC_50_ = 5.25 ± 0.74 µM, respectively [38]. 

Baldisserotto et al. obtained benzimidazole derivatives endowed with phenolic hydroxy groups and a hydrazone moiety in previous research [39]; however, in the current study their anti-melanoma potential was investigated [40]. Compound **(11)** showed the most promising activity against the melanoma cell line Colo-38 with an IC_50_ = 0.50 ± 0.12 µM. The compound with the highest activity bore the 4-diethylamino-2-hydroxyphenyl group. The activity of derivatives increased in the given series of N1 substituents: 2,3,4-(OH)_3_-phenyl < 2,4-(OH)_2_-phenyl < 2,5-(OH)_2_-phenyl < 3-OH-4-OMe-phenyl < 2-OH-4-N(Et)_2_-phenyl for the most potential agents.

### 2.3. Imidazothiazole Derivatives

Imidazo[2,1-b]thiazole is a fused heterocycle that is of great importance in medical chemistry due to its wide range of biological activity. Its derivatives exert antitumor activity by inhibiting tubulin polymerization, kinases, or inducing apoptosis [41]. Imidazothiazole derivatives with anti-melanoma activity are presented in Figure 4.

Ammar et al. obtained novel imidazo[2,1-b]thiazole derivatives bearing m-nitrophenyl group at the 6-position [42]. In the present study, the authors modified previously reported inhibitor structures of 3-fluoro-substituted scaffolds based on imidazo[2,1 -b]thiazole [43]. Electron-withdrawing properties were left with concomitant introduction of additional H-bond acceptors to improve binding to the active site BRAF^V600E^. Compound **(12)** exhibited the highest activity against BRAF^V600E^ with an IC_50_ = 0.020 ± 0.0003 µM. In the cell line study, compound **(13)** had the greatest activity against melanoma cell lines LOX IMVI, M14, MALME-3M, MDA-MB-435, and UACC-257 with GI_50_ = 2.57 µM, 2.72 µM, 2.00 µM, 2.55 µM, and 2.54 µM, respectively. Compound **(14)** was the most active against melanoma cell lines SK-MEL-2, SK-MEL-28, SK-MEL-5, and UACC-62 with GI_50_ = 2.04 µM, 2.63 µM, 0.82 µM, and 1.16 µM, respectively. SAR analysis indicated that compounds bearing a propyl linker between the terminal arylsulfonamide group and the pyrimidine ring at the 5-position of the imidazo[2,1-b]thiazole ring possessed higher activity than those with the ethyl linker [42].

Ammar et al. obtained novel NH_2_-based imidazothiazole derivatives based on the structural features of Dabrafenib [44]. Dabrafenib is a BRAF serine/threonine kinase inhibitor approved for the treatment of metastatic melanoma as monotherapy or in combination with Trametinib [45]. The novel derivatives bore an amine substitution on the terminal phenyl ring, a sulfonamide group on the side of the chain, and a terminally substituted phenyl ring. The aim of the syntheses was to obtain new BRAF^V600E^ kinase inhibitors. Compound **(15)** exhibited the highest activity against BRAF^V600E^ with an IC_50_ = 1.20 nM. The authors noted that the activity can be related to the propylene linker and the 4-methoxyphenyl ring in the side chain, which provided excellent conditions for high affinity and formation of a solid complex with BRAF. A single dose (10µM) an in vitro cytotoxic assay indicated that, in the melanoma cell line UACC-62, Compound **(16)** possessed 100% inhibition. Moreover, the SAR study indicated that compounds bearing the aforementioned propyl linker exhibited an overall higher inhibitory activity on the UACC-62 melanoma cell line than compounds bearing an ethylene linker [44].

Abdel-Maksoud et al. obtained novel imidazo[2,1-b]thiazole derivatives [46]. The main goal of the syntheses was to discover new inhibitors of BRAFV^600E^ and BRAF^WildType^. Compound **(17)** was the most active against BRAF^V600E^ with an IC_50_ = 0.98 ± 0.012 nM and Compound **(18)** exhibited the highest activity against BRAF^WildType^ with an IC_50_ = 2.7 ± 1 nM. Surprisingly, the compounds possessing the highest activity in the BRAF^V600E^ and BRAF^WildType^ kinase enzyme assay did not show the highest activity in the cell assay. Compound **(19)** exhibited the broadest spectrum of activity against melanoma cell lines MALME-3M, M14, SK-MEL-2, SK-MEL-5, and UACC-62 with an IC_50_ = 0.036 μM, 0.174 μM, 2.26 μM, 0.348 μM, and 0.058 μM, respectively. Compound **(41)** showed the most promising activity against melanoma cell lines MDA-MB-435, SK-MEL-28, and UACC-257 with an IC_50_ = 0.381 µM, 0.176 μM, and 0.170 μM. Furthermore, the immunoblot test indicated that **(20)** could inhibit the phosphorylation of MEK and ERK, making it an excellent potential agent for melanoma [46].

### 2.4. Quinoline Derivatives

Quinoline contains a benzene ring fused with pyridine on two adjacent carbon atoms. Its derivatives exert antitumor activity by inhibiting tubulin polymerization, DNA repair, angiogenesis, key enzymes for tumor growth, and the ability to intercalate DNA [47,48,49,50]. Quinoline derivatives with anti-melanoma activity are presented in Figure 5.

El-Damasy et al. obtained novel 2-anilinoquinoline derivatives [51]. The aim of the synthesis was to introduce a phenyl ring with different lipophilic and steric substituents as an alternative for huge aromatic groups of 2-arylamides with anilinoquinolines, while maintaining the amide linker. Compound **(21)** exhibited the most promising activity against melanoma cell line MDA-MB-435 with GI_50_ = 0.878 µM, as compared to the reference standard Imatinib, which possessed about 20-fold less activity with GI_50_ = 17.91 µM. Compound **(22)** showed the highest and the broadest spectrum of activity against many lines of melanoma cells, such as LOX IMVI, MALME-3M, M14, and SK-MEL-28 with GI_50_ = 0.870 µM, 0.611 µM, 0.668 µM, and 0.846 µM, respectively, as compared to the previously mentioned standard with GI_50_ = 18.11 µM, 16.33 µM, 19.28 µM, and 14.62 µM, respectively. Moreover, it exhibited microtubule polymerization stabilizing effects comparable to paclitaxel. The authors noted that the key to the antitumor activity was the presence of 3-trifluoromethylphenyl group substituted with cyclic amines, which resulted in an enhanced steric effect. Experimental results were confirmed by molecular docking study [51].

Elbadawi et al. synthesized novel 4-alkoxy-2-aryl-6,7-dimethoxyquinolines [52]. Compound **(23)** exhibited the most promising activity against melanoma cell lines LOX IMVI, MDA-MB-435, and SK-MEL-5 with an IC_50_ = 0.116 µM, 0.709 µM, and 0.247 µM, respectively; **(24)** was active against melanoma cell line M14 with an IC_50_ = 0.327 µM; and **(25)** was active against melanoma cell line UACC-62 with an IC_50_ = 0.628 µM. Derivatives bearing a trifluoromethyl group in the para position of the phenyl ring possesses higher activity in regard to the chlorine atom in the same position. The authors noted that for the 4-position of the quinoline ring, the presence of the propyl linker between the oxygen atom and the aryl group is crucial for the activity. Within this aryl group, the activity increased in the given series: 4-hydroxypiperidine > morpholine > piperidine > pyrrolidine, and for the 2-position of the quinoline ring, the activity increased in the given series: 2-thiophenyl < 3-thiophenyl < 2-furanyl < phenyl [52].

Albadari et al. obtained novel derivatives with high anti-melanoma activity, based on the hydroxyquinoline scaffold [53]. The previous study indicated that the 8-hydroxyquinoline moiety is crucial for antiproliferative activity [42] Compound **(26)** exhibited the most promising activity against the multi-drug resistant melanoma line MDA-MB-435/LCC6MDRI with an IC_50_ = 0.2 ± 0.0 µM, as well as **(27)** against melanoma cell lines A375 and MDA-MB-435 with an IC_50_ = 0.5 ± 0.1 µM for both lines. Both compounds **(28)** and **(29)** possessed the same activity against melanoma cell line RPMI7951 with an IC_50_ = 0.5 ± 0.1 µM. The results indicated that the phenyl, 4-isoquinoline, 3-thiophene, and bromine groups seem to be crucial for the activity [53].

Manikala et al. obtained new chalcone-tethered quinoline derivatives [54]. Compound **(30)** exhibited the most promising activity against melanoma cell line A375 with an IC_50_ = 0.34 µM, as compared to the reference standard Adriamycin, which possessed about 16-fold less activity with an IC_50_ = 5.51 µM. The most active compound bore the nitro group in the para position of the phenyl ring. The activity of the derivatives increased in the given series of phenyl substituents: 4-bromo < 3,4,5-trimethoxy < 3-nitro < 4-nitro for the most potential agents [54].

### 2.5. Pyrazolopirimidines Derivatives

Among the N-fused heterocycles, pyrazolopyrimidine is one of the favored scaffolds due to its synthesis and pharmacological importance. Its derivatives exert antitumor activity mainly as kinase inhibitors [55]. Pyrazolopirimidines derivatives with anti-melanoma activity are presented in Figure 6.

Li et al. obtained new pyrazolo [1,5-a]pyrimidine derivatives [56]. In their previous research, the authors synthesized derivatives of 4-substituted methoxybenzoyl-arylthiazoles [57] and 2-aryl-4-benzoyl-imidazoles [58,59] with significant antitumor activity. However, it turned out that the presence of the unstable carbonyl group between the imidazole and phenyl ring is susceptible to metabolic reduction. Therefore, by means of a fusion strategy, this group was incorporated into the pyrimidine ring. The pyrazolo [1,5-a]pyrimidine scaffold plays the role of an imidazole and ketone bioisostere. The 3,4,5-trimethoxyphene group was retained related to its pharmacophoric nature for antitumor activity. Compound **(31)** showed the highest activity against B16-F10 tumor cells with an IC_50_ = 0.021 ± 0.002 µM, which is about four-fold more active than the reference standard colchicine with an IC_50_ = 0.087 ± 0.006 µM and is comparable to paclitaxel with an IC_50_ = 0.021 ± 0.002 µM. Compound **(31)** inhibited tubulin polymerization in an in vitro study and induced cell cycle arrest in the G2/M phase. These results were confirmed by an in vivo study in the B16-F10 murine melanoma model where tumor growth was inhibited without apparent toxicity. The SAR study of various aromatic groups indicated that the most favored for activity are the presence of electron-donating groups in the phenyl ring. The highest activity was shown by the compound substituted with the methyl group in the para position of the phenyl ring. The key to the activity is the presence of 3,4,5-trimethoxyphenyl moiety at the 7-position of the pyrazolo[1,5-a]pyrimidine ring, and for the 2-position, the activity increased in the given series: 3-amino-4-methoxyphenyl < 6-indole < 5-(1-methyl-1H-indole) < 4-methylphenyl for the most potential agents. Moreover, the introduced modifications increased the metabolic stability of the compound [56].

Ibrutinib is an FDA-approved Bruton’s tyrosine kinase (BTK) inhibitor used in the treatment of leukemias [60]. Moreover, its promising activity against solid tumors has been shown [61,62,63,64]. Related to the widespread drug resistance and significant side effects [65,66,67], it is necessary to search for new drugs with increased effectiveness and lower toxicity. As a result, modifications to the structure of ibrutinib are of great interest to scientists looking for anticancer drugs [68]. Based on the literature data, the thiourea moiety was introduced, which ensures selectivity, effectiveness, and favorable physicochemical parameters [69,70,71,72,73]. Compound **(32)** exhibited the highest activity against melanoma cell line B16 with an IC_50_ = 6.07 ± 1.06 µM [74]. Furthermore, it arrested cells in the G1 phase and promoted cell apoptosis and autophagy in melanoma cell line B16. In an in vivo study, **(32)** significantly inhibited the growth of melanoma in mice [74].

### 2.6. Indole Derivatives

Indole consists of a benzene ring fused at the 2-position and 3-position of the pyrrole ring. Its derivatives exert antitumor activity as inhibitors of tubulin polymerization and topoisomerase and as inducers of apoptosis, aromatase, and kinase inhibitors [75]. Indole derivatives with anti-melanoma activity are presented in Figure 7.

Dinavahi et al. synthesized new 2,3-dioxo-2,3-dihydroindol-1-yl derivatives [76]. The aim of the syntheses was to obtain inhibitors of aldehyde dehydrogenase (ALDH), which is overexpressed in various neoplasms [77,78,79,80]. Compound **(33)** showed the highest activity against melanoma cell line UACC 903 with an IC_50_ = 3 ± 0.6 µM, and **(34)** against melanoma cell line 1205 Lu with an IC_50_ = 2.1 ± 0.6 µM. One of the mechanisms of action of the proposed compounds is to induce the activity of reactive oxygen species, lipid peroxidation, and accumulation of toxic aldehydes, as well as cell cycle arrest in the G2/M phase and apoptosis [76].

### 2.7. Sesquiterpene Lactone Derivatives

Sesquiterpenes are natural terpenoids consisting of a 15-carbon skeleton, isolated from plants of the Asteraceae family. The sesquiterpene may be hydrocarbon-based, or it may contain functional moieties, such as a lactone group. Sesquiterpenes have well-established anticancer activity [81,82]. Sesquiterpene lactone derivatives with anti-melanoma activity are presented in Figure 8.

Penthala et al. obtained novel sesquiterpene lactone analogs [83]. Compound **(35),** bearing a disubstituted phenyl ring with a methoxy group at the 3-position and 5-position, exerted sole activity against melanoma cell line UACC-257 with GI_50_ = 0.66 µM. Compound **(36),** bearing a methoxy group in the para position of the phenyl ring exhibited the broadest spectrum of activity against melanoma cell lines LOX IMVI, MALME-3M, M14, SK-MEL-2, SK-MEL-5, and UACC-62 with GI_50_ = 0.20 µM, 0.19 µM, 0.42 µM, 0.27 µM, 0.32 µM, and 0.29 µM, respectively [83].

### 2.8. Heterotricyclic Derivatives

Three-ring condensed heterocycles are a valuable scaffold with antitumor activity in medical chemistry [84,85]. Its derivatives with anti-melanoma activity are presented in Figure 9.

Chang et al. obtained 1-arylamino-3-aryloxypropan-2-ol derivatives as potent anti-melanoma agents [86]. The basis for the drug scaffold design were studies confirming the anti-melanoma activity of β-adrenergic receptor antagonists (β-blockers), commonly used in cardiac diseases [87,88,89,90,91,92,93,94]. The authors performed an initial screening of popular β-blockers in three human malignant melanoma cell lines (SK-MEL-5, SK-MEL-28, and A375) using MTT assay. Carvedilol exhibited the highest activity on all cell lines with an IC_50_ = 13.73 ± 0.20 μM, 15.55 ± 0.11 μM and 13.01 ± 0.78 μM, respectively. Carvedilol is a structure based on the carbazole scaffold, which is one of the privileged pharmacophores with antineoplastic activity [95,96]. Compound (**37)** exhibited the most promising activity against the SK-MEL-5 melanoma cell line with an IC_50_ = 2.69 ± 0.18 µM; however, lower activity against the remaining cell lines was not subjected to further detailed studies [86].

Gomaa et al. obtained the novel 2,3-dihydropyrazino[1,2-a]indole-1,4-dione derivatives, as a potential dual BRAF/EGFR inhibitors [97]. The key factor for the therapy seems to be the simultaneous inhibition of the epidermal growth factor receptor (EGFR), which may be activated as a result of BRAF inhibition. This leads to progressive tumor proliferation [98]. The previously obtained compounds [99] were modified in the C10 atom by extending the alkyl group in order to improve their binding affinity. Compound **(38)** exhibited the most promising activity against BRAF^V600E^ with an IC_50_ = 45 ± 5 nM and melanoma cell line LOX-IMVI with an IC_50_ = 1.02 ± 0.02 µM. The SAR study indicated that the presence of phenethyl in C2 is crucial for the activity. This modification provided flexibility and allowed the benzyl group to expand hydrophobic interactions. Oxygen atom in C4 was responsible for hydrogen bond formation with Cys532 in BRAF^V600E^. The proliferative activity increased in the given series: morpholinyl > piperidyl > 2-methylpyrrolidinyl > dimethylamino for the para position of the phenylethyl tail. The derivative bearing the chlorine atom in C8 and an ethoxymethyl group at C10 shows the highest activity [97].

Patinote et al. obtained novel imidazo[1,2-a]quinoxalines derivatives [100]. The aim of the syntheses was to optimize the previously reported structure [101,102] so as to separate the antitumor effect from the interactions of tubulin, which is also present in normal cells. The results of molecular modeling allowed the design of the second generation of imiqualine bearing the 3,4-dihydroxyphenyl moiety in the 1-position. Compound **(39)** exhibited the highest activity against melanoma cell lines A375, ME WO, A2058, and IPC 298 with an IC_50_ = 3 nM, 3 nM, 4 nM, and 40 nM, respectively, as compared to the reference standard Vemurafenib with an IC_50_ = 139 nM, 283 nM, 425 nM, and 13,000 nM, respectively. In the heterotopic mouse model of the heterotopic human melanoma xenograft, A375 cells were inoculated subcutaneously and the proposed antitumor compound was shown to be strongly correlated with a low mitotic index and lower invasion capacity [100].

Chitti et al. obtained novel 7-(5-((substituted-amino)-methyl)-thiophen-2-yl)-spiro[chroman-2,4′-piperidin]-4-one hydrochloride analogues [103]. The spirochromanone core was selected on the basis of previously reported derivatives [104]. Compound **(40)** did not exhibit the highest activity against melanoma cell line B16F10 with an IC_50_ = 13.15 ± 1.9 µM, but it possessed the highest selectivity index (SI = 13.37) for them, which made it the one possessing the highest potential. The purulent compound induced apoptosis by arresting the cell cycle in the G2 phase. The proposed compound was the only one to have a simple methylamine moiety; the remaining compounds had longer and more branched or cyclic chains. This may indicate that the methylamine group was crucial for the selectivity of spirochromanone derivatives for melanoma cell line B16F10 [103].

Aly et al. obtained a novel series of paracyclophanyl-dihydronaphtho[2,3-d]thiazoles [105]. Compound **(41)** showed the most promising activity against melanoma cell line SK-MEL-5 with an IC_50_ = 0.81 ± 0.03 µM, as compared to the reference standard Dinaciclib, which exhibited about seven-fold less activity with an IC_50_ = 5.97 ± 0.25 µM). Paracyclophane/thiazole conjugates bearing the naphthoquinone moiety with the allyl substituent in the thiazole ring exhibited the highest activity, which may be related to the improvement in binding affinity to the target protein and to the increase in flexibility [105].

### 2.9. Jaspine Derivatives

Jaspine B is a cyclic anhydrophytosphingosine isolated from sea sponges, Pachastissamina sp., and Jaspis sp. It is a natural sphingolipid with a broad spectrum of activity, including anticancer. Its derivatives with anti-melanoma activity are presented in Figure 10.

Yang et al. obtained 2-epi-jaspine B analogs [106]. Compound **(42)** turned out to be about two-fold more active against melanoma cell line A375 with an IC_50_ = 0.70 ± 0.09 µM, as compared to standard Cisplatin with an IC_50_ = 1.3 ± 0.11 µM. The proposed compound induced apoptosis of the A375 melanoma cell line by arresting the cell cycle in the G1 phase. The most active derivative bore the tetrahydro-2H-pyran-4-yl moiety [106].

Novotná et al. obtained new aza-derivatives of Jaspine B [107]. The basis for the use of the Jaspine B scaffold was its demonstrated ability to induce apoptosis in melanoma cell line SK-MEL-28 [108]. Compound **(43)** exhibited the highest activity against melanoma cell line BLM with an IC_50_ = 31.01 ± 0.89 µM/L, as compared to standard Cisplatin with an IC_50_ = 32.18 ± 1.29 µM/L [107].

### 2.10. Cinnamic Acid Derivatives

Cinnamic acid exerts a broad spectrum of biological activity [109]. There are many studies on anti-melanoma activity [110,111,112]; new derivatives are presented in the Figure 11.

Vale et al. obtained new cinnamic acid ester derivatives [113]. The basis for the syntheses was the confirmed anticancer activity of cinnamic acid and its derivatives against melanoma [110,111,112]. The results of the melanoma cell line B16-F10 viability assay exhibited that compound **(44)** possessed the highest activity with an IC_50_ = 60.28 µM. Moreover, at a concentration of 25 μM, the proposed compound significantly inhibited the migration of the B16-F10 melanoma cell line. The trypan blue method showed that **(44)** reduced the proliferation of the B16-F10 melanoma cell line at all concentrations tested (6.25, 12.5, 25, and 50 μM) after 24h of treatment. The authors suggested that the probable mechanism may be related to the function of p53, a transcription factor responsible, among others, for inducing cell sugar arrest, promoting apoptosis, activating oncogenes or inhibiting cell proliferation [114]. Biological studies also indicated that **(44)** disturbed cell invasion, adhesion, colonization, and polymerization of actin. The high activity of the proposed compound is probably related to the addition of two bromine atoms to the double bond of the cinnamic acid structure, which had a favorable effect on bioavailability and interaction with the biological target [113].

Romagnoli et al. obtained new cinnamic acid derivatives linked to arylpiperazines [115]. Compound **(45)** showed the most promising activity against melanoma cell line A375 with GI_50_ = 16.1 ± 1.5 µM. The most active compound bore a chlorine atom and a fluorine atom at the 3-position and 4-position, respectively, of the phenyl ring connected to piperazine, and the amino and methoxy groups also at the 3-position and 4-position of the second phenyl ring, respectively [115].

### 2.11. Thiosemicarbazone and -zide Derivatives

Thiosemicarbazones and thiosemicarbazides are scaffolds with a privileged place in medical chemistry. Their derivatives exert a wide spectrum of activity, including anticancer activity [116,117]. Its derivatives with anti-melanoma activity are presented in Figure 12.

In our lab, we also obtained a series of compounds as potential agents for melanoma. Pitucha et al. synthesized the copper complexes of thiosemicarbazones [118] based on their confirmed anticancer activity [119,120,121]. Compound **(46)** showed the most promising activity against melanoma cell line G361 an IC_50_ = 135.64 ± 7.01µM, making it about three-fold more potent than the standard Dacarbazine (DAC) with an IC_50_ = 425.98 ± 4.74 µM; **(47)** against melanoma cell line A375 with an IC_50_ = 26.05 ± 1.75 µM, making it about 15-fold more active than the DAC with an IC_50_ = 412.77 ± 7.08 µM; and **(48)** against melanoma cell line SK-MEL-28 with an IC_50_ = 46.13 ± 2.74 µM, making it about eight-fold more active than the DAC with an IC_50_ = 370.12 ± 9.46 µM. More detailed studies on **(47)** and **(48)** indicated a possible mechanism of oxidative damage to DNA as a result of induction of reactive oxygen species (ROS) [118].

Kozyra et al. obtained new 2,4-dichlorophenoxyacetic acid hydrazide thiosemicarbazide derivatives [122]. We based our current research on previously confirmed anticancer activity of this scaffold [123]. Compound **(49)** turned out to be the most active against melanoma cell line G-361 with an IC_50_ = 99±4 µM. The proposed compound bore an iodine atom in the para position of the second phenyl ring [122]. The activity of the derivatives increased in the given series of substituents: 2-bromophenyl < napht-1-yl < 2-iodophenyl < 4-methylthiophenyl < 4-iodophenyl for the most potential agents.

### 2.12. Other Derivatives

The remaining derivatives, which could not be easily divided depending on the chemical structure, are shown in the Figure 13.

Based on the previously described approach, Chang et al. designed agent **(50)**, which possessed activity against melanoma cell lines SK-MEL-28 and A375 with an IC_50_ = 3.30 ± 0.06 μM and 1.98 ± 0.10μM, respectively [82]. Moreover, compound **(50)** inhibited G2/M phase cell arrest. Immunofluorescent staining possessed a disruption of the microtubule network upon exposure to (**50)**, leading to apoptosis and cell arrest, and indicated that tubulin could be a potential molecular target for **(50)**. This was confirmed by a molecular docking study [86].

Razmienė et al. obtained the series of N-aryl-2,6-diphenyl-2H-pyrazolo[4,3-c]pyridin-7-amines [124]. The aim of the syntheses was to obtain compounds whose activity increased after irradiation with visible blue light (414 nm). Compound **(51)** was most active against melanoma cell line G361, possessing an EC_50_ (the concentration of a drug that gives half-maximal response) = 3.5 µM, 1.6 µM, and 0.9 µM in cells irradiated with 1,5, and 10 J/cm^2^, respectively. The photodynamic potential of the proposed compound was also confirmed by the study on human HaCaT keratinocytes, which were about two-fold less sensitive to it. The likely mechanism of action is to induce the formation of reactive oxygen species in cells and damage DNA, leading to apoptosis [124].

Cui et al. obtained new pyrrolo[2,1-f][1,2,4]triazine derivatives [125]. The basis for the research was to obtain dual inhibitors, both targeting snail and histone deacetylase (HDAC). Snail is a transcription factor [126] and HDAC is involved in histone modification [127]. Related to the strong connection between the two pathways, the creation of double inhibitors enable a more effective therapy against cancer. With the highest activity against the above-mentioned targets, Compound **(52)** was also examined against melanoma cell line MDA-MB-435 with GI_50_ = 0.0361 µM. The most active derivative bore a fluorine atom in the para position of the phenyl ring beyond the amine group in the ortho position and a 5-methyl-1H-pyrazol-3-ylamino moiety in the 4-position pyrrolo[2,1-f][1,2,4]triazin-2-yl [125].

Hassan et al. obtained new pyrazoline derivatives [128]. An in vitro study of a single dose test (10-5 µM), determining the percent growth inhibition, indicated that the proposed Compound **(53)** possessed 98.46% growth inhibition of the SK-MEL-5 melanoma cell line. The most active derivative bore the 4-methoxyphenoxy and 3,4,5-trimethoxyphenyl moiety, which indicated the high potential of the electron-withdrawing group [128].

Gagné-Boulet et al. obtained novel derivatives of 1-(4-(phenylthio)phenyl)imidazolidin-2-one [129]. The imidazolidin-2-one scaffold was selected based on previous studies, and the sulfonate moiety from previous studies was replaced with a thioether linker [130]. Compound **(54)** exhibited the highest activity against melanoma cell line M21 with an IC_50_ = 0.2 ± 0.1 µM. The proposed compound arrested the progression of the cell cycle in the G2/M phase and induced the disruption of the cytoskeleton. The N,N’-ethylene-bis(iodoacetamide) (EBI) detection assay indicated the interference of the colchicine binding site on the microtubules by **(54)**. Moreover, **(54)** showed low toxicity of chicken embryo. The SAR study indicated that the key to the activity is the substituent on the terminal phenyl group. The most active compound bore the chlorine atom substituted in the meta position. Furthermore, the presence of a thioether linker as well as the imidazolidin-2-one moiety is also important for the activity [129].

Zhang et al. obtained novel nitric oxide-releasing derivatives of triptolide [131]. Triptolide was first isolated from Tripterygium wilfordii Hak. f. It exhibited a broad antitumor spectrum, inter alia, in a mouse model, it inhibited the growth of open-bone xenografts, among others, by melanoma cell line B16, and it inhibited the metastasis of the B16F10 melanoma cell line to the spleen and lungs [132]. The interest in nitric oxide as an anticancer factor stems from its ability to induce apoptosis, sensitize cancer cells to chemotherapy, or inhibit metastasis [133,134,135]. Therefore, furoxan, as a nitric oxide donor, was introduced at the 14-position of tryptolide. Compound **(55)** possessed the most promising activity against murine B16F10 melanoma cell with an IC_50_ = 0.021 ± 0.03 µM/L, which is comparable to triptolide with an IC_50_ = 0.005 ± 0.002 µM/L. Further studies involving a xenograft mouse model of melanoma established by the subcutaneous inoculation of B16F10 showed that the proposed compound inhibited the growth of melanoma at low doses (0.3 mg/kg). Moreover, the SAR study indicated that the length of the linker connecting the furoxan ring with the triptolide scaffold is inversely proportional to the activity [131].

Abdel-Maksoud et al. obtained novel derivatives based on the pyrrolo[2,3-b]pyridine scaffold [136]. It has a privileged nature and is present in the approved and preclinically tested inhibitors of B-RAF [137,138,139,140]. Compound **(56)** showed the most promising activity against BRAF^V600E^ with an IC_50_ = 0.080 ± 0.003 µM. The derivative bearing a bromine atom in the para position of the phenyl ring possessed the highest activity. For the most potential agents, activity increased in a given series of substituents 3-(4-substituted-phenyl)urea: trifluoromethyl < methyl < bromine [136].

Pałkowski et al. obtained novel 1-methyl 3-octyloxymethylimidazolium derivatives carrying various anionic moieties [141]. The basis for the syntheses were the few scientific reports on the anticancer activity of ionic liquids [142,143]. Compound **(57)** exhibited the most promising activity against melanoma cell line B16F10 with IC_50_ = 1.01 × 10^−2^ µM/L. The most active derivative bore the thymoloxyacetic anion [141].

Barbarossa et al. obtained new thalidomide derivatives [144]. Despite its infamous history, thalidomide has been successfully repositioned for cancer treatment [145]. Currently, efforts are focused on optimizing the phthalimide core of thalidomide, which is a pharmacophore, in order to reduce side effects [146,147]. Based on the high antitumor activity of compounds correlated with thalidomide from previous studies [148], a series of derivatives were obtained based on the privileged phthalimide core connected directly to a differently substituted aromatic ring with the N-terminus of the phthalimide moiety. The most promising compound turned out to be **(58)**, against melanoma cell line A2058 with an IC_50_ = 15.37 ± 0.7 µM. The immunofluorescence and the tubulin polymerization test indicated that it interfered with the dynamics of microtubules by inhibiting tubulin polymerization. This was confirmed by the results of molecular docking. The TUNEL assay showed the potential to induce apoptosis in melanoma cell line A2058. For the most potential agents, the activity increased in a given series of substituents: 4-phenoxyphenyl < 2-chlorophenoxyphenyl < 2-methylphenoxyphenyl [144].

Geng et al. obtained novel diaryl derivatives [149]. Compound **(59),** bearing 3 bromine atoms in the phenyl ring, turned out to be the most potent agent against melanoma cell line A375 with an IC_50_ = 2.95 ± 0.23 µM, as compared to standard Cisplatin with an IC_50_ = 3.75 ± 0.62 µM. The proposed compound leads to apoptosis through the inducing G2/M phase arrest of the A375 melanoma cell line. The results were confirmed by a transplanted tumor nude mouse model [149].

Piechowska et al. obtained a series of hybrid compounds with tropinone and thiazole rings [150]. The choice of such a combination of both groups was based on previous research results [151]. The most promising activity against melanoma cell line B16-F10 with an IC_50_ = 1.51 ± 0.14 µM was exhibited by **(60)**, as compared to standard Chlorambucil with an IC_50_ = 2.92 ± 0.10 µM. The most active derivative bore the 4-iodophenyl substituent [150].

Remigante et al. investigated the effect of the known BK activator NS-11021 **(61)** on the viability, proliferation, and migration of IGR39 cells of the melanoma cell line [25]. The authors maintained that the above-mentioned effects are not specifically dependent on the opening of the BK channels. Compound **(61)**, in addition to direct activation of BK channels, also resulted in an increase in intracellular calcium levels. The latter effect is likely due to the activation of the calcium-permeable conductivity and it is this elevation in intracellular calcium levels that may be responsible for inhibiting the viability, proliferation, and migration of IGR39 cells of the melanoma cell line. The latter discovery provides a new potential direction for new potential agents with anti-melanoma activity [25].

## 3. Conclusions

This article provides a summary overview of the latest potential melanoma agents from 2020-2022. SAR analysis was discussed for selected structures. Therefore, the present review may be useful for rational drug design with anti-melanoma activity.

## Figures and Tables

**Figure 1 ijms-23-06084-f001:**
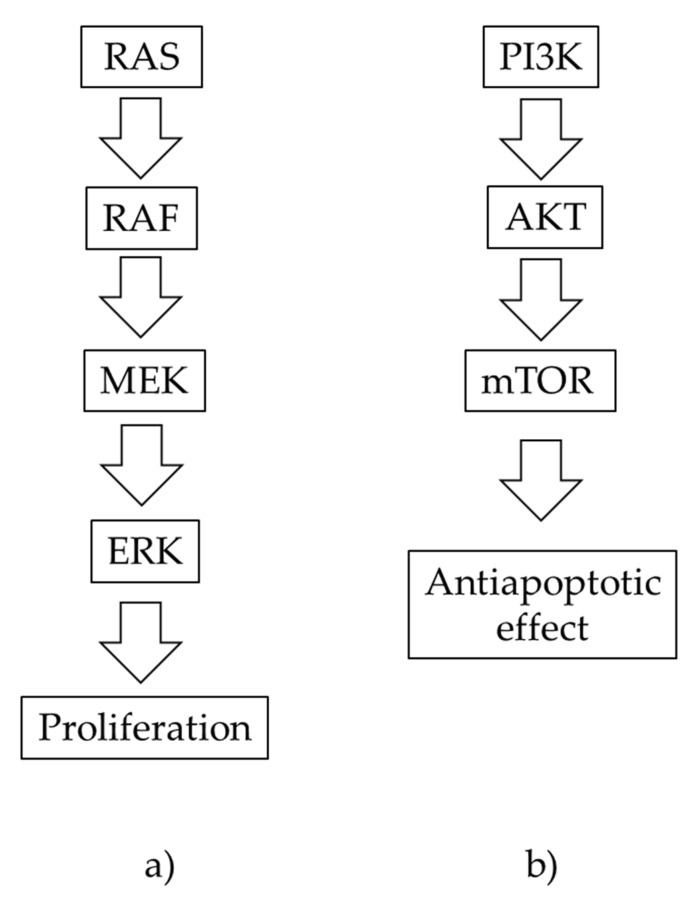
A simplified scheme of the most studied signaling pathways for the development of melanoma: (**a**) MAPK pathway, (**b**) PI3K pathway.

**Figure 2 ijms-23-06084-f002:**
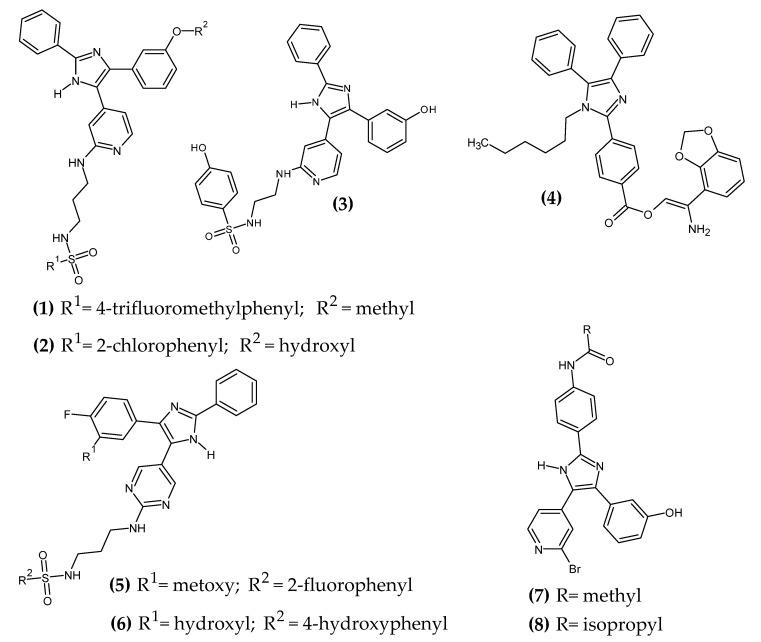
Imidazole derivatives with anti-melanoma activity.

**Figure 3 ijms-23-06084-f003:**
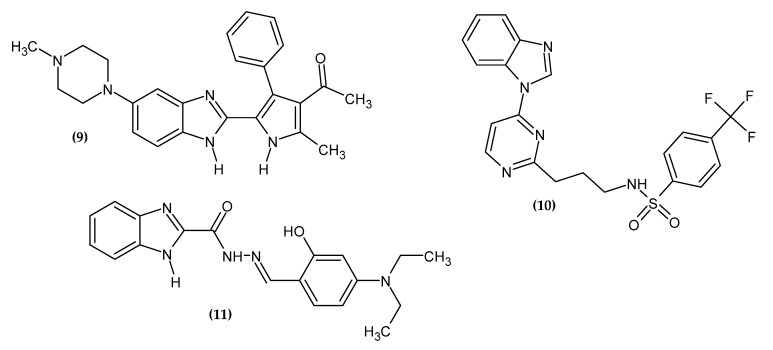
Benzimidazole derivatives with anti-melanoma activity.

**Figure 4 ijms-23-06084-f004:**
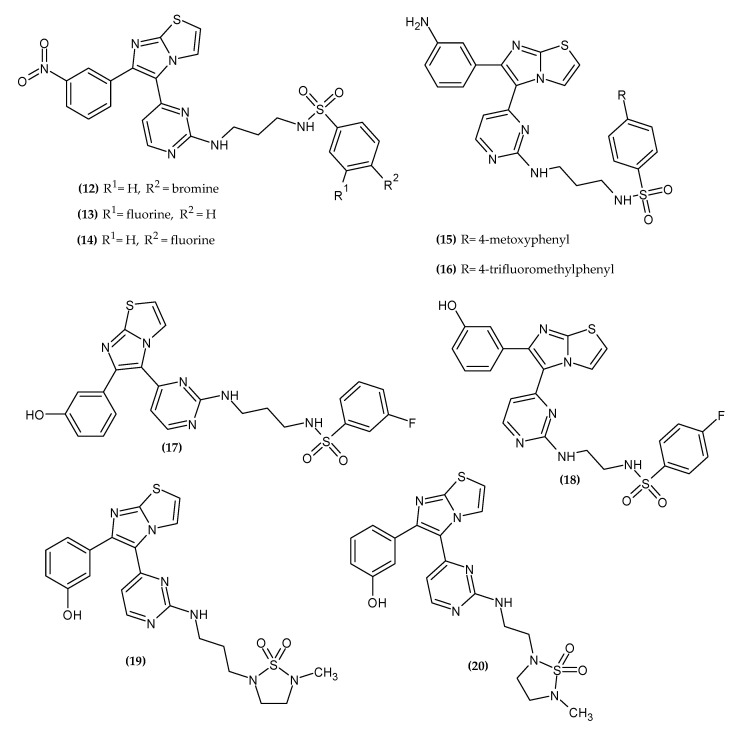
Imidazothiazole derivatives with anti-melanoma activity.

**Figure 5 ijms-23-06084-f005:**
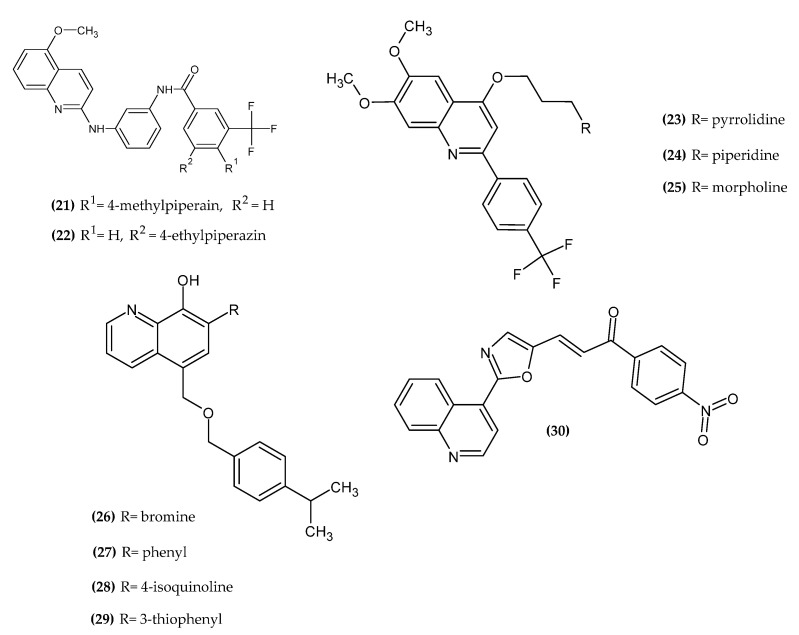
Quinoline derivatives with anti-melanoma activity.

**Figure 6 ijms-23-06084-f006:**
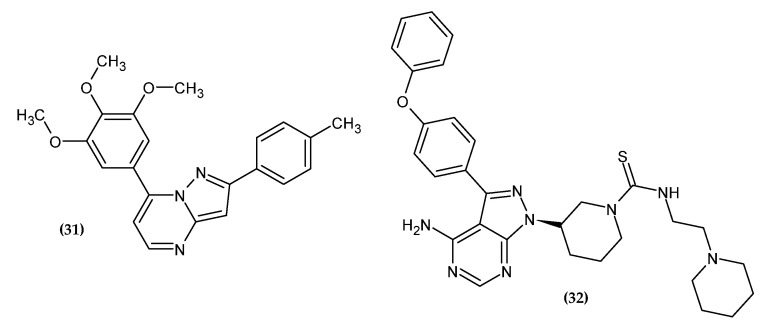
Pyrazolopirimidine derivatives with anti-melanoma activity.

**Figure 7 ijms-23-06084-f007:**
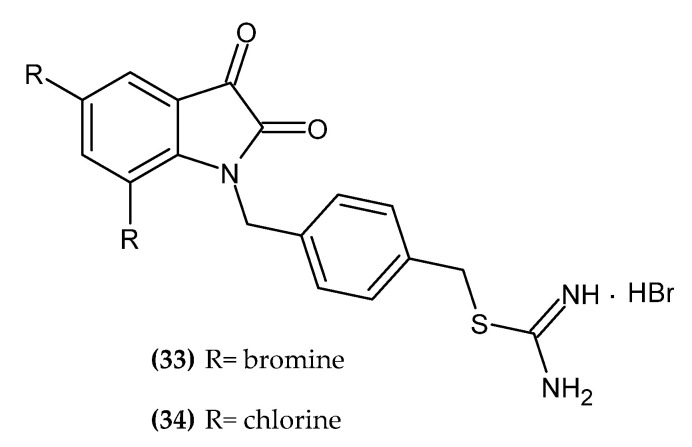
Indole derivatives with anti-melanoma activity.

**Figure 8 ijms-23-06084-f008:**
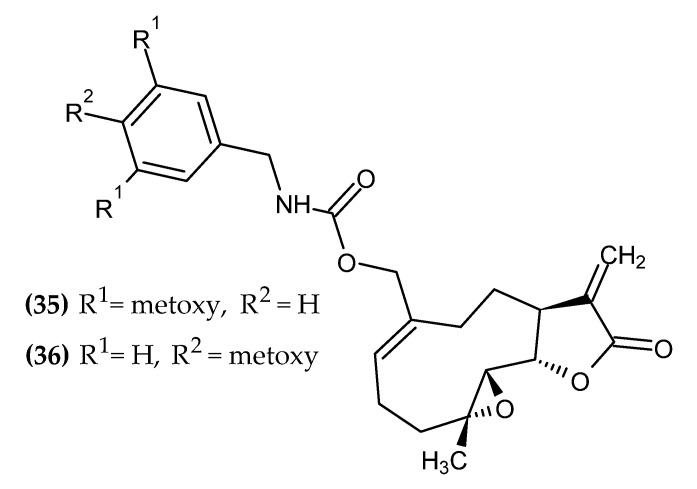
Sesquiterpene lactone derivatives with anti-melanoma activity.

**Figure 9 ijms-23-06084-f009:**
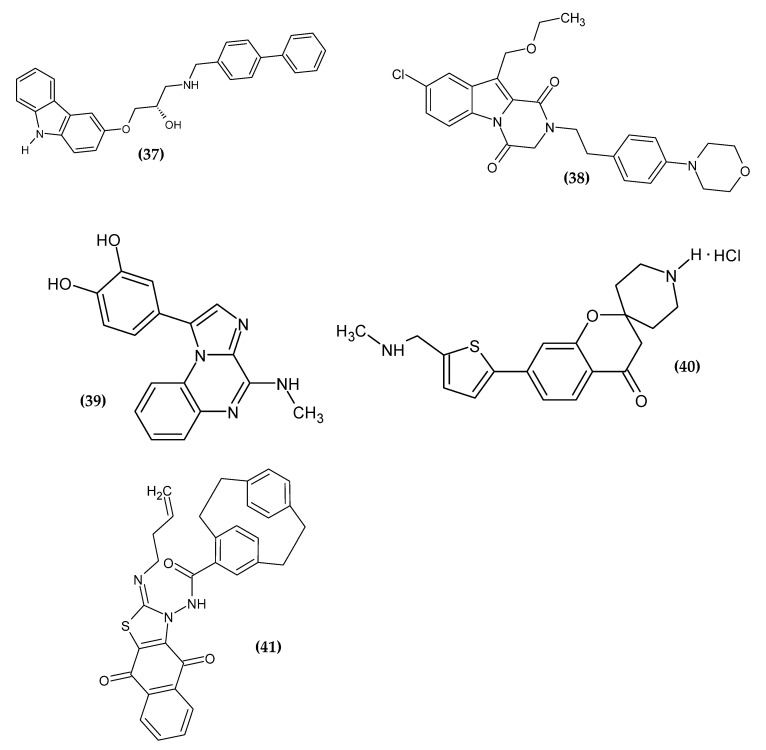
Heterotricyclic derivatives with anti-melanoma activity.

**Figure 10 ijms-23-06084-f010:**
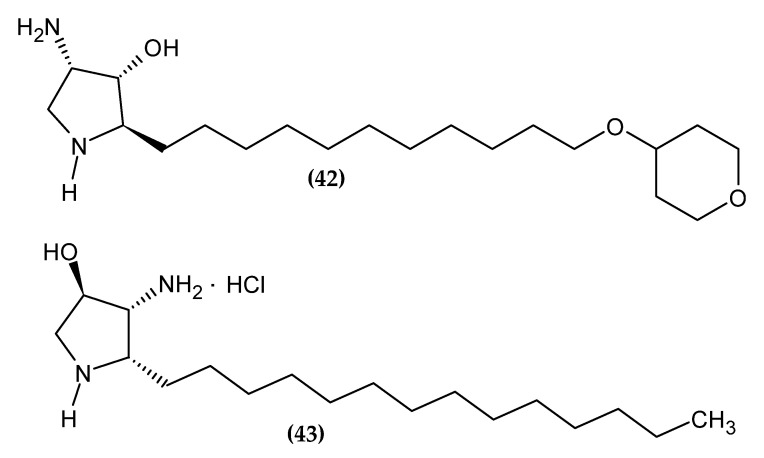
Jaspine B derivatives with anti-melanoma activity.

**Figure 11 ijms-23-06084-f011:**
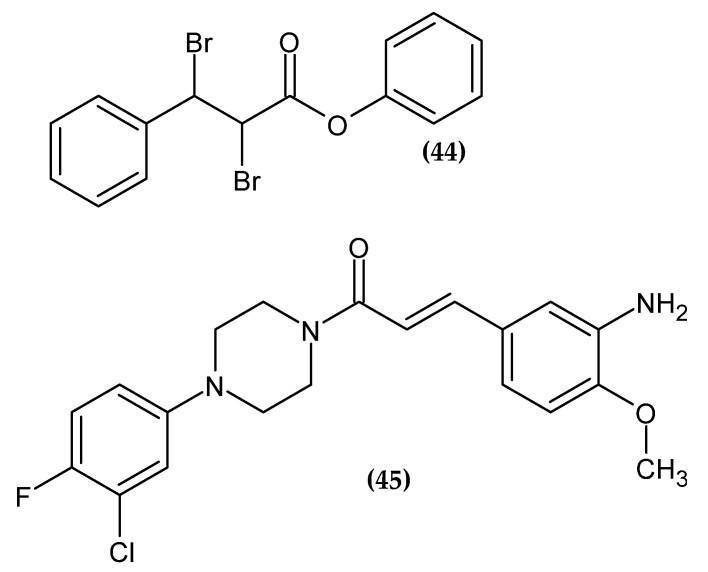
Cinnamic acid derivatives with anti-melanoma activity.

**Figure 12 ijms-23-06084-f012:**
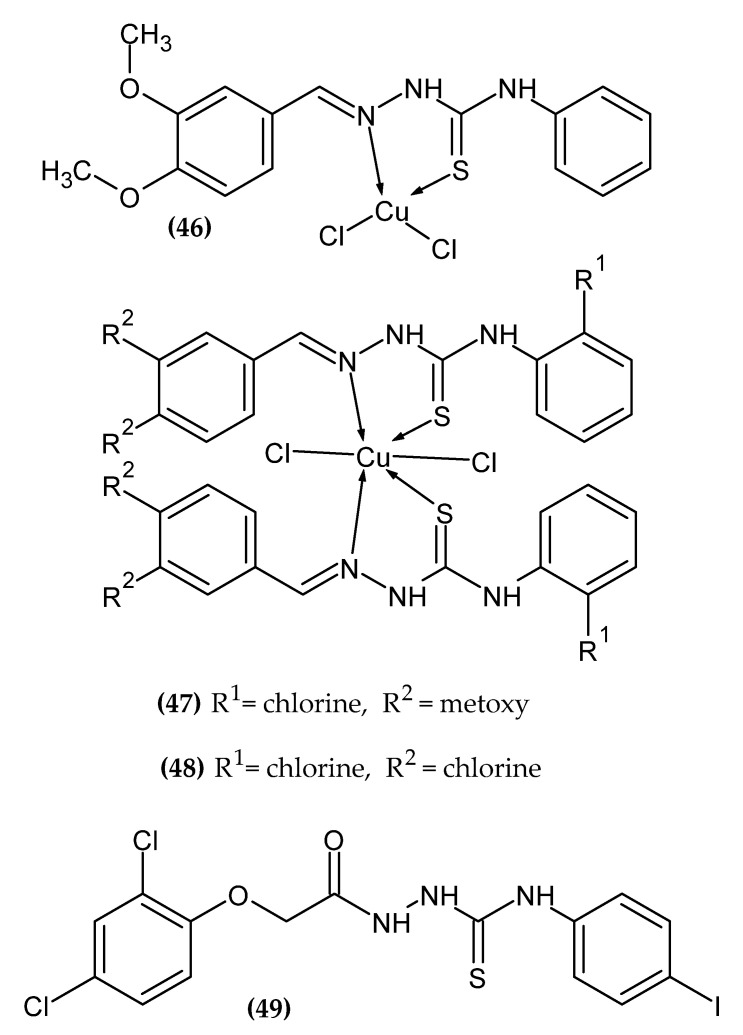
Thiosemicarbazone and -zide derivatives with anti-melanoma activity.

**Figure 13 ijms-23-06084-f013:**
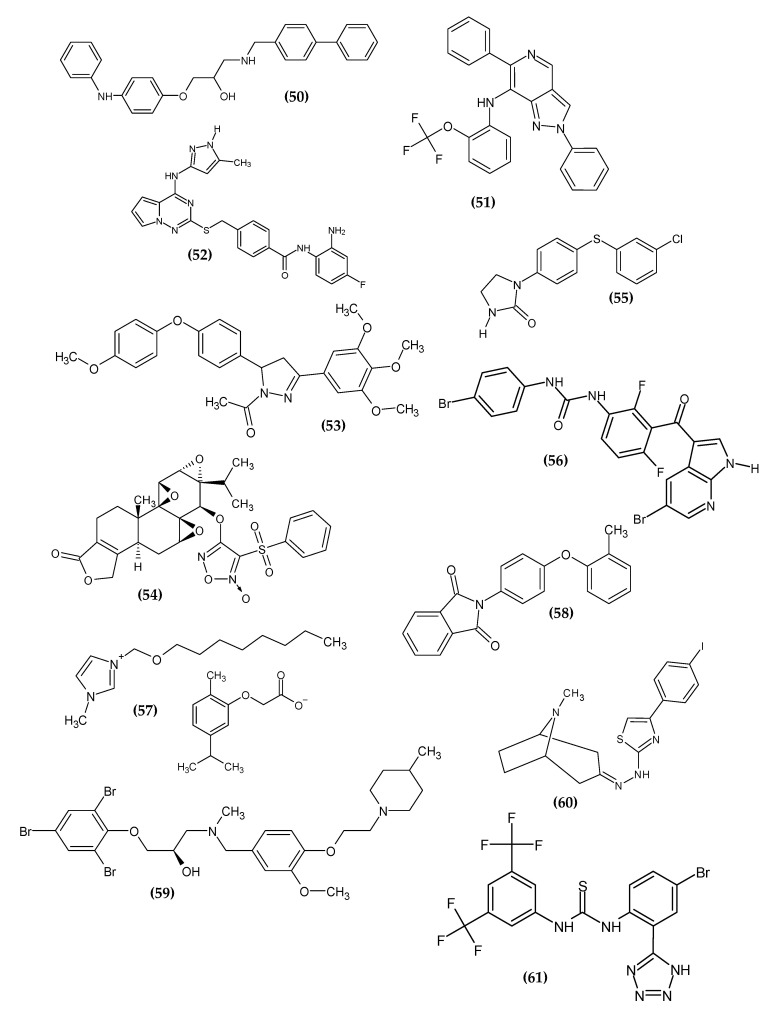
Other derivatives with anti-melanoma activity.

## Data Availability

Not applicable.

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
