# Peer review of "New Potential Agents for Malignant Melanoma Treatment—Most Recent Studies 2020–2022"

_ijms, 2022, doi:10.3390/ijms23116084_

Round 1
Reviewer 1 Report
The manuscript "New potential agents for Melanoma Malignant treatment - most recent studies 2020-2022" is a comprehensive and exhaustive review, worthy to be published in IJMS. Nevertheless, its organization is poor, or, at least, not very evident; it would be more readable if subdivisions are added to ease readers their search of information. The following concerns need to be revised:
Major concerns:
1) The review is comprehensive and exhaustive. But a line, a "story", is missed. Which one is the logical order of the citations/compounds reviewed? Why Authors have not included any subdivision to make the article more organized and ordered?
It would be desirable to include any logical subdivision that improves the readability of the manuscript. For instance, classifying the compounds in function of the main ring present in the structure; into carbocycles, five-membered rings, six-membered rings, fused rings... Another possibility is to classify the compounds by mechanism of action. The first would be a medicinal chemistry approach; the second a more biological one.
Without this the review is a good summary of the field, but without an index that help the reader to search his/her interests.
Minor concerns
2) Why Conclusions are section "5" when the previous one is the section "2"? Where are sections 3 and 4?
3) Please draw figures always after its first citation, never before, as it can be confusing to the readers.
4) Having 39 figures is good - an exhaustive art work - but has a small inconvenience. Each compound / family of compounds has its/their respective figure. Perhaps it would be good to draw the figures in small groups of 4-8 figures. It is not for the space (MDPI journals have no limitation), it is for enabling a quick comparison between similar figures, which help the readers, and in special the Medicinal Chemists, to compare different structures and seek similarities faster.
5) Chemical notation: when an element is cited in the name of the compound [example: (1H-imidazol)], the element should be in italics (1H-imidazol).
6) The correct way of using the "et al." expression is: "AAAAA et al.". With italics, and not "et all.". Please correct along all the manuscript.
7) English is generally good and readable, but it needs a careful check to correct typos and small mistakes.
Reviewer 2 Report
This review focuses on the most recent synthesized anti-melanoma agents from 2020-2022. The scientific collect is very interesting, however, some aspects, as indicated below, should be addressed before the document can be considered for publication in this journal. This version of the manuscript is not enough complete.
Minor revision:
-English language and style are fine/minor spell check required.
-I suggest to review the style of the manuscript according to the guidelines of the journal.
-The authors should divided in more subparagraphs the section 2.
Major revision:
Recent research has revealed that ion channels can be important players in development, progression, and therapy resistance in melanoma. Specifically, calcium-dependent potassium channels have emerged as regulators of carcinogenesis, thus introducing possible new therapeutic strategies in the fight against melanoma. As potassium channels are involved in various pathophysiological conditions, the biophysical mechanisms underlying their regulation have been extensively studied, and potassium channel activators (NS-1102, BMS-19011 and/or NS-19504) are sought as novel avenues to the treatment of diseases, including cancer. Based on this evidence, I suggest to add a new section in the main text in which the ion channels can serve as a promising pharmacological target in melanoma treatment, and their dysregulation might serve as a marker for melanoma prediction. Thus, I suggest to add some recent papers (DOI: 10.3390/cancers13236144, DOI: 10.1186/s12885-020-07071-1).
Round 2
Reviewer 1 Report
The manuscript "New potential agents for Melanoma Malignant treatment - most recent studies 2020-2022" has improved significantly since the last version and now is suitable for publication in its current form. Congratulations for your excellent work.
Reviewer 2 Report
The authors have completely soddisfated the issues of the reviewer. Thus, this manuscript cab be accepted in the present form to be publish in this journal.